# Growth across life course and cardiovascular risk markers in 18-year-old adolescents: the 1993 Pelotas birth cohort

Romina Buffarini,[1] María Clara Restrepo-Méndez,[2] Vera Maria Silveira,[3] Helen D Gonçalves,[1] Isabel O Oliveira,[4] Ana Maria Menezes,[1] Maria Cecília Formoso Assunção[5]

[1]Post-graduate Program in Epidemiology, Federal University of Pelotas, Pelotas, Brazil
[2]MRC Integrative Epidemiology Unit, Population Health Sciences, Bristol Medical School, University of Bristol, Bristol, UK
[3]Clinical Medical Department, Faculty of Medicine, Federal University of Pelotas, Pelotas, Brazil
[4]Department of Physiology and Pharmacology, Federal University of Pelotas, Pelotas, Brazil
[5]Department of Nutrition, School of Nutrition, Federal University of Pelotas, Pelotas, Brazil

**Correspondence to**
Dr Romina Buffarini;
romibuffarini@gmail.com

## ABSTRACT

**Objective** To evaluate the association between growth trajectories from birth to adolescence and cardiovascular risk marker levels at age 18 years in a population-based cohort. In order to disentangle the effect of weight gain from that of height gain, growth was analysed using conditional weight relative to linear growth (CWh) and conditional length/height (CH).

**Design** Prospective study.

**Setting** 1993 Pelotas birth cohort, Southern Brazil.

**Participants** Individuals who have been followed up from birth to adolescence (at birth, 1, 4, 11, 15 and 18 years).

**Primary outcome measures** C-reactive protein (CRP), total cholesterol (TC), LDL-cholesterol (LDL-C), HDL-cholesterol (HDL-C), triglycerides (TGL), systolic and diastolic blood pressure (SBP and DBP), body mass index (BMI) and waist circumference (WC).

**Results** In both sexes, greater CWh at 1 year was positively associated with BMI and WC, whereas greater CWh at most age periods in childhood and adolescence predicted higher CRP, TC, LDL-C, TGL, SBP, DBP, BMI and WC levels, as well as lower HDL-C level. Higher CH during infancy and childhood was positively related with SBP in boys and girls, and with BMI and WC only in boys.

**Conclusion** Our study shows that rapid weight gain from 1 year onwards is positively associated with several markers of cardiovascular risk at 18 years. Overall, our results for the first year of life add evidence to the 'first 1000 days initiative' suggesting that prevention of excessive weight gain in childhood might be important in reducing subsequent cardiovascular risk.

## Strengths and limitations of this study

► Population-based sample birth cohort; data collected prospectively over 18 years since birth with high follow-up rates.
► Availability of several anthropometric and biological markers of cardiovascular risk.
► Use of conditional growth to examine highly correlated measurements, and assessment of the separate contributions of linear growth and relative weight gain.
► Use of subsample in one of the follow-ups. Multiple hypothesis testing leads to increased type I error.
► Lack of anthropometric data at 2 years is a limitation, since there is evidence suggesting that the hazards of rapid weight gain appear particularly after the first 2 years of life.

resistance[4] and cardiovascular diseases.[5 6] However, the critical age interval (first year of life or middle childhood) for accelerated growth (weight and height changes) that leads to the development of chronic conditions remains controversial.[5 7 8] Evidence suggests that accelerated weight gain during infancy is associated with an adverse metabolic risk profile in adulthood.[7 9 10] In addition, it has been found that rapid weight gain increases the risk of metabolic disturbances when it occurs during childhood, as a result of a more rapid development of body fat mass.[11] The definition of this critical period is crucial to inform health policies.

Studies on growth trajectories throughout life course and adult outcomes have examined weight gain without any distinction between the weight gain relative to height and linear growth. Dissimilar consequences of these measures have been found,[12 13] which are relevant for healthcare policy makers when developing programmes for cardiometabolic risk prevention. In this study we aimed to assess the associations between size at birth (weight and length), conditional

## INTRODUCTION

Metabolic and cardiovascular diseases are important public health problems that cause enormous costs in terms of human and economic resources and are responsible for high rates of morbidity and mortality in most regions of the world.[1]

Growth trajectories throughout life course, including the fetal period, may have effects on the cardiovascular risk profile later in life.[2] It is well established in the literature that low birth weight (a marker of fetal growth restriction) is a risk factor for type 2 diabetes,[3] insulin

relative weight (CWh) and conditional length/height (CH) at ages 1, 4, 11, 15 and 18 years and cardiovascular risk markers (C-reactive protein (CRP), lipid profile, body mass index (BMI) and waist circumference (WC)) in adolescents aged 18 years who are participants in the 1993 Pelotas birth cohort study.

## METHODS

The city of Pelotas, a medium-sized city with nearly 330 000 inhabitants, is located in southern Brazil. In 1993, all mothers who delivered a newborn in any of the five hospitals of the city and who resided in the urban area were invited to participate in a birth cohort study (99% of all births). Data were collected on 5249 live births and only 16 individuals refused to participate. The cohort participants have been followed up at different time points thereafter. All visits were carried out by trained interviewers and fieldwork team members. Further details of the methodology have been published previously.[14]

This study included information from six follow-ups: perinatal and at ages 1, 4, 11, 15 and 18 years. The 1-year and 4-year follow-ups were conducted only in a subsample of the original cohort. The same children were the target population (n=1460) for both follow-ups which consisted of all low birth weight children (n=510) plus a random sample of 20% of children who were not born with low birth weight (n=950). The response rates for each follow-up were 99.6% (n=5249), 93.4% (n=1363), 87.2% (n=1273), 87.5% (n=4452), 85.7% (n=4349) and 81.4% (n=4106), respectively (see online supplementary file 1). More details of each follow-up are described elsewhere.[15] A diagram with a description of the cohort follow-ups used in this study is available in online supplementary file 1. Household visits were performed in every wave except for the 18-year-old visit which took place at the university research clinic where interviews, physical examinations and collection of biological samples were carried out.[16]

The protocol study was approved by the School of Medicine Ethics Committee of the Federal University of Pelotas. All participants or their legal representatives voluntarily signed a consent letter prior to participation in each follow-up. Verbal consent was provided in the perinatal phase.

### Assessment of outcomes

We examined the following cardiovascular risk markers measured at the 18-year follow-up: CRP, total cholesterol (TC), HDL-cholesterol (HDL-C), LDL-cholesterol (LDL-C), triglycerides (TGL), systolic blood pressure (SBP), diastolic blood pressure (DBP), BMI and WC.

Venous blood samples were collected regardless of fasting status, left at room temperature for 30 min and then centrifuged for 15 min at 2000 g. Serum aliquots were stored at −80°C until analysis. Blood samples were not taken in pregnant or suspected pregnant participants (n=59). Lipids were measured using an automatic enzymatic colorimetric method in a biochemistry analyzer

(BS-380 Mindray; Shenzhen Mindray Bio-Medical Electronics, China). CPR was measured by turbidimetric immunoassay also using the BS-380 analyzer. Blood pressure was recorded in the seated position using a calibrated digital wrist monitor (Omron HEM-629, Beijing, China). Measurements were taken at the start and at the end of the visit and the mean of the two measurements was used in the analysis.

Current BMI was calculated by dividing weight in kilograms by height squared in metres. Weight was measured using a scale coupled to BodPod equipment (Life Measurement, Concord, California, USA) and height was obtained with standardised techniques using a wall–mounted stadiometer (SECA 240; Seca, Birmingham, UK). Waist circumference was measured with a fibreglass tape at the narrowest point of the torso.[17]

### Assessment of exposures

Birth weight and length were recorded in the maternity hospitals at delivery. In the subsequent follow-ups the measurements were conducted at the participant's home and at the research clinic in the 18-year follow-up. On each occasion, weight and length or height were measured by trained field workers using standard equipment and protocols.

We studied growth patterns in several life periods: infancy (from birth to 1 year), early and mid-childhood (1–4 and 4–11 years old, respectively) and early and late adolescence (11–15 and 15–18 years old, respectively). For each age interval the separate effects of weight gain and linear growth were examined using CWh and CH proposed by Adair et al.[12] CWh takes into account current height and preceding weights and lengths or heights, and CH takes into account prior weight and length or height measures but not current weight.[12]

To calculate these conditional measures we computed sex-specific internally derived z-scores for weight and length or height at each follow-up. We then regressed the z-score size measurements (weight or height) at a given age on z-scores at all previous measurements. The conditional measure is represented by the standardised residuals derived from the regression and indicates the deviation from the individual's expected measures, in view of his or her previous growth and the average growth of the cohort members. This could be interpreted as a measure of relatively faster or slower weight or length/height change over a period of time. For example, an adolescent with a positive CWh value from 11 to 15 years put on more weight compared with his or her previous weights and length/heights and weights and length/heights of all cohort participants. As the conditional variables are uncorrelated, they can be included in a multiple regression model without breaking any assumption of collinearity.[18]

### Statistical analyses

We first described the outcomes by means (SD) (geometric means for asymmetric data: CRP and TGL); t-tests were

used to estimate mean differences between sexes. To assess the association between each outcome and conditional growth we used linear regression and P values were obtained by Wald's test. The outcomes were standardised to allow direct comparisons of the regression coefficients. We adjusted for the following confounding factors: family income (in minimum wages), maternal education (completed years of schooling) at birth, self-reported skin colour (white, black or others), breastfeeding (total duration of breastfeeding in months) and maternal smoking during pregnancy (yes/no). Unadjusted and adjusted coefficients and statistical significance of associations did not differ markedly with the inclusion of confounders in our models, so we presented only adjusted results. We included a term of interaction by sex in the model which provided statistical evidence of interaction in most of the associations, therefore analyses are shown stratified by sex. Analyses were performed using Stata 12.1 (Stata Corp, College Station, Texas, USA).

## RESULTS

At a mean (SD) age of 18.5 (0.25) years, 4106 adolescents were evaluated. CWh and CH data across infancy, childhood and adolescence were available for 957 participants (23.4% of the 18-year-old follow-up). Our main analysis samples consisted of those cohort members who had complete data on exposures, confounders and at least one outcome. Therefore, our sample sizes were 917 for blood examinations and 946 for blood pressure, BMI and WC.

Mean levels of SBP and WC were higher in boys than in girls. On the other hand, girls had higher values of TC, HDL-C, LDL-C and TGL compared with boys. No differences were shown for DBP and BMI by sex (table 1).

Mean outcome values of the main analysis samples were compared with mean outcome values of all participants who had information on blood tests, blood pressure and anthropometric measures. Only a few differences were observed between both samples. Boys included in the main analyses had lower CPR and higher HDL-C values compared with boys who were evaluated at the 18-year follow-up. Among girls, those included in the main analyses showed higher LDL-C than the total girls who were examined at the 18-year follow-up (see online supplementary file 2). The proportions of boys and girls were slightly different in the two samples, with boys slightly under-represented in the main analysis sample compared with the sample comprising all participants who had information on blood tests, blood pressure and anthropometric measures (about 47.6% vs 49.8%). Comparing those included in the main analysis sample with those excluded, we observed statistical evidence of differences in mean values of TGL and WC among boys (see online supplementary file 3).

Tables 2 and 3 show the association of size at birth, CWh and CH with cardiovascular risk markers at 18 years of age.

**Table 1** Mean (SD) outcomes at 18 years of age in the main analyses samples stratified by sex: 1993 Pelotas birth cohort*

| Outcomes | Boys (n=438) | Girls (n=479) | P value |
|---|---|---|---|
| C-reactive protein (mg/L)† | 0.64 (3.10) | 1.35 (3.92) | <0.001 |
| Total cholesterol (mg/dL) | 151.31 (24.46) | 172.29 (30.35) | <0.001 |
| HDL cholesterol (mg/dL) | 52.78 (8.75) | 59.50 (10.54) | <0.001 |
| LDL cholesterol (mg/dL) | 83.55 (18.46) | 97.14 (25.42) | <0.001 |
| Triglycerides (mg/dL)† | 70.81 (1.46) | 74.79 (1.44) | 0.03 |
| Systolic blood pressure (mm Hg)‡ | 130.35 (11.56) | 115.40 (10.04) | <0.001 |
| Diastolic blood pressure (mm Hg)‡ | 70.54 (8.29) | 69.71 (7.83) | 0.12 |
| Body mass index (kg/m$^2$)‡ | 23.00 (4.25) | 23.58 (5.05) | 0.05 |
| Waist circumference (cm)‡ | 77.51 (9.71) | 73.74 (10.32) | <0.001 |

Data are arithmetic mean (SD) unless otherwise indicated.
*Main analysis sample includes individuals with complete data on all growth measures, all confounders and at least one outcome.
†Geometric mean (SD).
‡Boys n=447; girls n=499.
HDL, high density lipoprotien; LDL, low density lipoprotein.

CRP levels showed a positive association with CWh during childhood and adolescence (1–4 years, 4–11 years and 11–15 years in both sexes and 15–18 years only in girls). The association between CRP and weight gain appeared to be stronger and with an increasing trend across age periods in girls. Positive associations between CRP and CH between 1 and 4 years were found in both sexes (tables 2 and 3).

Overall, the lipid profile was associated with CWh gain during childhood and adolescence, especially in boys. CWh in early childhood (1–4 years) was positively associated only with TGL, and CWh in children aged 4–11 years was positively associated with LDL-C and TGL and negatively associated with HDL-C (table 2). CWh during early and late adolescence (11–15 years and 15–18 years) was positively related with TC, LDL-C and TGL and negatively related with HDL-C (table 2). In girls, HDL-C was inversely associated only with CWh in mid-childhood (4–11 years) and both early and late adolescence (11–15 years and 15–18 years), whereas TC and LDL-C and TGL showed positive associations with CWh in late adolescence (table 3).

Findings for SBP were similar for both sexes. However, DBP showed different patterns between sexes. Higher CWh during early and mid-childhood (1–4 years and 4–11

**Table 2** Association of conditional relative weight and conditional height with cardiovascular risk markers at 18 years of age in boys: 1993 Pelotas birth cohort

| | CRP | TC | HDL-C | LDL-C | TGL | SBP | DBP | BMI | WC |
|---|---|---|---|---|---|---|---|---|---|
| **Cardiovascular risk markers** | | | | | | | | | |
| **Conditional weight (CWh)** | | | | | | | | | |
| 0–1 year | −0.00 (−0.04 to 0.04) | 0.07 (−0.01 to 0.14) | 0.03 (−0.04 to 0.10) | 0.07 (−0.01 to 0.15) | 0.01 (−0.01 to 0.03) | 0.04 (−0.04 to 0.112) | 0.04 (−0.04 to 0.13) | 0.26 (0.18 to 0.34) | 0.27 (0.19 to 0.35) |
| 1–4 years | 0.08 (0.03 to 0.13) | 0.03 (−0.05 to 0.12) | −0.03 (−0.11 to 0.05) | 0.05 (−0.03 to 0.11) | 0.04 (0.01 to 0.06) | 0.09 (0.01 to 0.17) | 0.05 (−0.04 to 0.15) | 0.43 (0.35 to 0.50) | 0.40 (0.33 to 0.48) |
| 4–11 years | 0.10 (0.05 to 0.14) | 0.06 (−0.01 to 0.13) | −0.08 (−0.15 to −0.01) | 0.09 (0.01 to 0.16) | 0.02 (0.01 to 0.04) | 0.08 (0.01 to 0.16) | 0.10 (0.01 to 0.19) | 0.48 (0.42 to 0.53) | 0.44 (0.38 to 0.50) |
| 11–15 years | 0.07 (0.02 to 0.12) | 0.13 (0.04 to 0.20) | 0.01 (−0.08 to 0.09) | 0.14 (0.07 to 0.23) | 0.04 (0.02 to 0.06) | 0.08 (0.01 to 0.15) | 0.08 (−0.01 to 0.17) | 0.33 (0.28 to 0.38) | 0.29 (0.24 to 0.36) |
| 15–18 years | 0.04 (−0.01 to 0.09) | 0.16 (0.08 to 0.23) | −0.11 (−0.19 to −0.04) | 0.18 (0.11 to 0.26) | 0.07 (0.05 to 0.09) | 0.12 (0.04 to 0.20) | 0.08 (−0.01 to 0.18) | 0.46 (0.44 to 0.47) | 0.47 (0.45 to 0.50) |
| **Conditional length/height (CH)** | | | | | | | | | |
| 0–1 year | 0.00 (−0.04 to 0.05) | −0.01 (−0.09 to 0.07) | −0.01 (−0.09 to 0.07) | −0.00 (−0.08 to 0.08) | −0.01 (−0.03 to 0.02) | 0.14 (0.06 to 0.23) | 0.08 (−0.02 to 0.18) | 0.11 (0.02 to 0.19) | 0.20 (0.11 to 0.29) |
| 1–4 years | 0.05 (0.01 to 0.11) | 0.06 (−0.02 to 0.14) | 0.03 (−0.05 to 0.11) | 0.05 (−0.04 to 0.13) | 0.02 (−0.01 to 0.04) | 0.09 (0.01 to 0.17) | 0.04 (−0.06 to 0.14) | 0.18 (0.09 to 0.26) | 0.24 (0.17 to 0.33) |
| 4–11 years | 0.00 (−0.05 to 0.04) | 0.06 (−0.03 to 0.12) | −0.08 (−0.15 to −0.01) | 0.07 (−0.01 to 0.15) | 0.02 (−0.00 to 0.05) | 0.13 (0.06 to 0.21) | 0.21 (0.12 to 0.30) | 0.13 (0.07 to 0.20) | 0.18 (0.11 to 0.26) |
| 11–15 years | −0.03 (−0.09 to 0.01) | −0.00 (−0.08 to 0.07) | 0.01 (−0.07 to 0.08) | 0.00 (−0.08 to 0.08) | −0.02 (−0.04 to 0.00) | 0.03 (−0.05 to 0.11) | 0.04 (−0.06 to 0.14) | −0.02 (−0.08 to 0.04) | 0.02 (−0.04 to 0.08) |
| 15–18 years | −0.00 (−0.05 to 0.05) | −0.10 (−0.17 to −0.02) | −0.03 (−0.11 to 0.04) | −0.08 (−0.15 to −0.00) | −0.02 (−0.04 to 0.00) | 0.07 (−0.01 to 0.14) | 0.02 (−0.08 to 0.11) | −0.06 (−0.10 to −0.01) | 0.02 (−0.03 to 0.08) |

Data are β (95% CI). The outcome variables were normalised. Regression coefficient (β) values were calculated with linear regression models and indicate the SD change in the outcome per SD change in the predictor.
All models were adjusted for smoking during pregnancy, breastfeeding, mother's education (years of schooling) and household wealth (in minimum wages) at birth and skin colour of the adolescent.
BMI, body mass index; CH, conditional height; CRP, C-reactive protein; CWh, conditional relative weight; DBP, diastolic blood pressure; HDL-C, high-density lipoprotein cholesterol; LDL-C, low-density lipoprotein cholesterol; SBP, systolic blood pressure; TC, total cholesterol; TGL, triglycerides; WC, waist circumference.

**Table 3**  Association of conditional relative weight and conditional height with cardiovascular risk markers at 18 years of age in girls: 1993 Pelotas birth cohort

Cardiovascular irsk markers

| | CRP | TC | HDL-C | LDL-C | TGL | SBP | DBP | BMI | WC |
|---|---|---|---|---|---|---|---|---|---|
| **Conditional weight (CWh)** | | | | | | | | | |
| 0–1 year | 0.03 (−0.05 to 0.12) | 0.01 (−0.10 to 0.10) | −0.04 (−0.13 to 0.05) | −0.01 (−0.10 to 0.10) | 0.01 (−0.03 to 0.06) | 0.06 (−0.00 to 0.13) | 0.03 (−0.06 to 0.12) | 0.25 (0.15 to 0.35) | 0.22 (0.13 to 0.31) |
| 1–4 years | 0.08 (−0.01 to 0.18) | −0.03 (−0.13 to 0.07) | −0.03 (−0.12 to 0.08) | −0.04 (−0.15 to 0.07) | 0.02 (−0.03 to 0.07) | 0.12 (0.05 to 0.18) | 0.12 (0.03 to 0.22) | 0.56 (0.47 to 0.65) | 0.45 (0.37 to 0.54) |
| 4–11 years | 0.11 (0.02; 0.20) | −0.04 (−0.13 to 0.06) | −0.15 (−0.24 to −0.07) | 0.03 (−0.07 to 0.13) | 0.00 (−0.02 to 0.02) | 0.10 (0.04 to 0.17) | 0.13 (0.04 to 0.22) | 0.62 (0.55 to 0.68) | 0.53 (0.47 to 0.59) |
| 11–15 years | 0.14 (0.06 to 0.23) | 0.00 (−0.09 to 0.10) | −0.18 (−0.26 to −0.09) | 0.06 (−0.05 to 0.15) | 0.03 (0.01 to 0.05) | 0.15 (0.09 to 0.21) | 0.11 (0.03 to 0.19) | 0.45 (0.40 to 0.50) | 0.40 (0.34 to 0.45) |
| 15–18 years | 0.22 (0.13 to 0.31) | 0.14 (0.04 to 0.24) | −0.10 (−0.19 to −0.02) | 0.20 (0.09 to 0.29) | 0.03 (0.01 to 0.06) | 0.16 (0.10 to 0.22) | 0.13 (0.05 to 0.22) | 0.51 (0.50 to 0.53) | 0.47 (0.43 to 0.49) |
| **Conditional length/height (CH)** | | | | | | | | | |
| 0–1 year | 0.01 (−0.08 to 0.12) | 0.08 (−0.01 to 0.17) | 0.16 (0.08 to 0.25) | 0.03 (−0.07 to 0.12) | −0.02 (−0.07 to 0.02) | 0.09 (0.02 to 0.16) | 0.01 (−0.08 to 0.10) | 0.01 (−0.08 to 0.10) | 0.08 (−0.00 to 0.17) |
| 1–4 years | 0.13 (0.04; 0.22) | 0.06 (−0.04; 0.17) | 0.11 (0.02; 0.20) | 0.01 (−0.09; 0.12) | 0.03 (−0.02; 0.08) | 0.11 (0.04; 0.18) | 0.11 (0.02; 0.21) | 0.13 (0.02; 0.23) | 0.22 (0.12; 0.31) |
| 4–11 years | −0.02 (−0.11; 0.08) | 0.05 (−0.04 to 0.15) | 0.03 (−0.05 to 0.12) | 0.04 (−0.06 to 0.14) | −0.00 (−0.02 to 0.02) | 0.08 (0.02 to 0.14) | 0.20 (0.11 to 0.29) | 0.12 (0.04 to 0.20) | 0.12 (0.04 to 0.20) |
| 11–15 years | −0.01 (−0.10; 0.08) | −0.02 (−0.11 to 0.08) | −0.07 (−0.15 to 0.02) | 0.02 (−0.09 to 0.12) | −0.01 (−0.03 to 0.02) | 0.08 (0.02 to 0.15) | 0.02 (−0.07 to 0.10) | −0.01 (−0.07 to 0.06) | 0.08 (0.01 to 0.13) |
| 15–18 years | −0.10 (−0.19 to −0.02) | 0.00 (−0.09 to 0.10) | −0.05 (−0.14 to 0.03) | 0.01 (−0.09 to 0.11) | 0.01 (−0.01 to 0.03) | −0.00 (−0.07 to 0.05) | 0.00 (−0.08 to 0.08) | −0.05 (−0.08 to −0.03) | 0.01 (−0.05 to 0.05) |

Data are β (95% CI). The outcome variables were normalised. Regression coefficient (β) values were calculated with linear regression models and indicate the SD change in the outcome per SD change in the predictor.
All models were adjusted for smoking during pregnancy, breastfeeding, mother's education (years of schooling), household wealth (in minimum wages) at birth and skin colour of the adolescent.
BMI, body mass index; CH, conditional height; CRP, reactive-C protein; CWh, conditional relative weight; DBP, diastolic blood pressure; HDL-C, high-density lipoprotein cholesterol; LDL-C, low-density lipoprotein cholesterol; SBP, systolic blood pressure; TC, total cholesterol; TGL, triglycerides; WC, weight circumference.

years) and through early and late adolescence (11–15 years and 15–18 years) was associated with higher SBP in both sexes, and with higher DBP only in girls. Regarding linear growth, conditional length gain during the first year of life and CH gain during early and mid-childhood (1–4 years and 4–11 years) was related with increased levels of SBP in both sexes, whereas CH gain during early and mid-childhood were related with increased levels of DBP only in girls (tables 2 and 3). Mean arterial pressure was also assessed. In general, the results are consistent with those shown for SBP and DBP in both sexes (see online supplementary file 4).

CWh in all age periods was associated with higher BMI and WC in boys and girls, with apparently larger coefficients after the first year of life (tables 2 and 3). CH between birth and 1 year of age was positively related with BMI and WC only in boys. CH throughout early and mid-childhood (1–4 years and 4–11 years) was positively related with BMI and WC in both sexes (tables 2 and 3). CWh was more strongly related to BMI and WC than linear growth. All the regression coefficients are shown in the original scales of outcomes (eg, mg/dL for lipid profile) in online supplementary file 5.

## DISCUSSION

Our results showed some differences in the relative contributions of weight gain and linear growth to cardiovascular risk. In general, we observed that greater CWh in childhood (1–4 years and 4–11 years) was positively associated with CRP, LDL-C, TGL, SBP, DBP, BMI and WC in boys, and with CRP, DBP, SBP, BMI and WC in girls (inverse associations with HDL-C in both sexes). Greater CWh in adolescence (11–15 years and 15–18 years) was positively associated with CRP, TC, LDL-C, TGL, SBP, BMI and WC and negatively associated with HDL-C in both sexes, and with DBP only in girls. Also, CWh during the first year of life was only related to BMI and WC in 18-year-old adolescents of both sexes. By contrast, associations between linear growth and the cardiovascular markers showed a less consistent pattern when compared with weight gain. Greater CH in infancy was associated with higher values of SBP in boys and girls, and with BMI and WC only in boys.

Previous evidence from low-, middle- and high-income countries showed that excessive weight gain predicts subsequent metabolic and cardiovascular diseases when it occurs in childhood, specifically after the second year of life.[12 13 19] Overall, our findings showing an association from early childhood (after the first year of age) onwards are consistent with this evidence. The association between weight gain throughout life and CRP concentrations in young adulthood was examined in the 1982 Pelotas birth cohort. In agreement with our analysis, the study showed that excessive weight gain at all age periods after the second year of life in both sexes was positively associated with CRP levels in 23-year-old participants of both sexes.[20] We also found that excessive weight gain during

childhood and adolescence was associated with increased total cholesterol, LDL-C and TGL and decreased HDL-C. Weight gain from birth to 4 years and lipid profile levels at 18 years were studied in the 1982 cohort and negative associations were found between excessive weight gain from 2 to 4 years and HDL-C, although the association was reduced after controlling for current BMI.[21] In a study using data from the ALSPAC cohort in England, the authors observed that greater BMI in mid-childhood predicted higher blood pressure (SBP and DBP) in 17-year-old adolescents.[22]

Fall *et al* assessed the relations between components of metabolic syndrome and weight gain at three periods from birth to 28 years of age when the outcomes were measured. In line with our findings, the study showed that greater weight gain in childhood (2–11 years) was associated with increased values of WC, SBP and TLG, while greater weight gain from 11 to 28 years predicted higher WC, TGL, total cholesterol and SBP and lower HDL-C. They also observed positive associations between rapid weight gain during the two first postnatal years and WC, SBP and TGL.[19]

In agreement with an analysis from the Vellore birth cohort in India,[23] we observed that rapid CWh in infancy (birth to 1 year) was positively associated with BMI and WC, but not with other cardiovascular markers. A meta-analysis assessing infant growth and subsequent obesity also showed that infants who grew more rapidly had a higher risk of developing obesity at later ages.[7] Studies that assessed the relation between growth and body composition showed that rapid weight gain in infancy was more related with fat-free mass than with fat mass in adulthood.[12 24] In contrast, rapid weight gain from childhood onwards generally is associated with accumulation of greater fat mass than with fat-free mass.[11 12 24–26] These findings may explain the positive associations of relative weight gain at all age periods with BMI, as this indicator does not distinguish body fat from fat-free mass.

In relation to linear growth, we observed that faster CH mainly in childhood was found to be positively associated with blood pressure (SBP and DBP), BMI and WC at 18 years of age in both sexes. A previous report with data from the same cohort examined the association between conditional growth at three different age ranges up to 4 years old, BMI and blood pressure at 15 years and found positive associations between conditional height from 1 to 4 years and both outcomes, although the association with SBP became insignificant when adjusted for BMI at 15 years.[27] Haugaard *et al* found a positive association between linear growth during childhood and SBP and WC at 8 years of age.[28] The previously mentioned Vellore birth cohort showed that rapid CH throughout the life course was positively associated with blood pressure and WC in young adults.[23] It is known that blood pressure is higher in taller people, which may be the result of an adaptation of the vascular function to perfuse a longer arterial tree. However, inverse relations between adult height and cardiovascular diseases have also been described,[29] which

suggests that this adaptation may not have pathological consequences.

The strengths of this study include a population-based sample and the availability of several anthropometric and biological markers of cardiovascular risk. Furthermore, our data have been collected prospectively since birth by trained staff with the use of standardised methods, reducing the susceptibility to misclassification. We also highlight the use of conditional growth to examine highly correlated measurements typical of longitudinal studies, and the assessment of the separate contributions of linear growth and weight gain relative to linear growth. Nevertheless, the assessment of relative weight gain does not distinguish between fat-free and fat mass gain.

We acknowledge some limitations of our study. First, our analyses were carried out only on those cohort participants with anthropometric data from several follow-ups, including subsamples. The potential impact of losses to follow-up on our results is difficult to assess. Second, we cannot rule out the possibility of random significant associations, as it is known that multiple hypothesis testing leads to increased type I error. Third, measurements at 2 years of age were not available in the 1993 Pelotas birth cohort and therefore we were not able to assess growth at this age point. The inclusion of 2 years of age in longitudinal studies assessing the associations between growth and cardiovascular risk is very important due to growing evidence that excessive relative weight gain denotes an increased risk for cardiovascular health when it occurs after 2 years of age. Although we do not have evidence of the impact of conditional growth at 2 years of age on later outcomes, our findings for the first year of life add to the evidence that improving nutrition during the 'first days of life' would result in long-term benefits on health, as greater infant weight gain and linear growth during this period have more benefits than risks for health, particularly improvement in capital human outcomes such as schooling and final achieved height.[12 13]

Although we studied a particular population (ie, children born in 1993 in the urban area of Pelotas, Brazil), we observed many similarities with the results of several studies, which suggests that our findings may be generalisable to populations in other low- and middle-income countries.

Given that cardiovascular risk can track from adolescence to adulthood,[30 31] a better understanding of the possible adverse effects of growth patterns at earlier stages in life can help to develop interventions aimed at preventing subsequent chronic diseases. Based on this study and other published evidence, we conclude that excessive weight gain from childhood onwards may have an adverse effect on cardiovascular health later in life. This reinforces efforts to inform strategies to avoid children putting on weight after their first 2 years of life for cardiovascular prevention. Evidence regarding linear growth throughout birth to late adolescence and subsequent cardiovascular risk remains unclear and needs further investigation.

**Acknowledgements** We are grateful to all the adolescents who took part in the Pelotas birth cohorts, and to the Pelotas teams including research scientists, interviewers, workers and volunteers.

**Contributors** RB, MCFA and MCR-M designed the study. RB performed the analysis. RB, MCR-M, MCFA and VMS contributed to the interpretation of the results and critical revision of the manuscript. HDG, AMM, IOO and MCFA participated in the design and conduct of the original cohort studies as well as in interpreting the results and reviewing the manuscript. RB wrote the manuscript. All authors contributed to and approved the final version.

**Funding** This article is based on data from the study "Pelotas Birth Cohort, 1993" conducted by Postgraduate Program in Epidemiology at Universidade Federal de Pelotas with the collaboration of the Brazilian Public Health Association (ABRASCO). From 2004 to 2013, the Wellcome Trust supported the 1993 birth cohort study. The European Union, National Support Program for Centers of Excellence (PRONEX), the Brazilian National Research Council (CNPq) and the Brazilian Ministry of Health supported previous phases of the study.

**Competing interests** None declared.

**Patient consent** Obtained.

**Ethics approval** The study and its protocols were approved by the School of Medicine Ethics Committee of the Federal University of Pelotas. All participants or their legal representatives voluntarily signed a consent letter (verbal consent was provided in perinatal phase) prior to participation in each follow-up.

**Provenance and peer review** Not commissioned; externally peer reviewed.

**Data sharing statement** Due to confidentiality restrictions related to the ethics approval for this study, no identifying information about participants may be released. As recipients, the authors were allowed to publish analytic results from the data, but not the data itself, due to confidentiality conditions.

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
