## [Reviewer comments · BMJ Open]

ARTICLE DETAILS

TITLE (PROVISIONAL)	Growth across life course and cardiovascular risk markers in 18 years old adolescents: the 1993 Pelotas Birth Cohort
AUTHORS	Buffarini, Romina; Restrepo-Méndez, Maria Clara; Silveira, Vera; Goncalves, Helen; Oliveira, Isabel; Menezes, Ana; Assuncao, M.C

VERSION 1 – REVIEW

REVIEWER	Chiara Di Gravio MRC Lifecourse Epidemiology Unit University of Southampton United Kingdom
REVIEW RETURNED	19-Sep-2017

GENERAL COMMENTS	The author looked at associations between growth trajectories from birth to adolescence and cardiometabolic risk markers at 18 years. Higher weight gain was positively associated with cardiometabolic markers, some associations between linear growth and cardiometabolic risk markers were also found. The statistical analysis is well-explained. Below are some comments/suggestions for the authors: Abstract: If space permit, include relevant estimates and 95% confidence interval in the results section and define TGL in the primary outcome section. Introduction: “...However, there is controversy about which age intervals of accelerated growth (weight and height changes) lead to the development of chronic conditions...” Which age intervals are considered in the literature and what controversy does the authors refer to? This statement needs more detail and clarification. “Dissimilar consequences of these measures have been found...” Provide references supporting this statement. Methods: In how many hospitals did women delivered? Were all the hospitals similar in term of population they cater to? It is unclear whether the 1,460 children mentioned are all of the children followed-up at 1 and 4 years, or if they are the random sample of 20% of the remaining population.
--

Suggest having different Ns in the text, one indicating how many low birth weight children were included, the other referring to the random sample of 20% of the remaining population.

Are the children randomly selected and included in the 1 year follow up, the same children studied at the 4 year follow up? Or was the sample randomly selected at both visit separately? This information becomes essential as the statistical methodology used (conditionals) is strongly affected by missing values.

Suggest adding the number of available measure at each follow up (together with the response rate already in the text). A flowchart of the cohort might help the reader in understanding the sample size at each follow-up.

Assessment of outcomes:
"The tests were not taken in pregnant or suspected pregnant." How many women?

Statistical Analyses:
Why using geometric means for CRP and TGL?

Why were the analysis stratified by sex?

Results:
"... a mean age of 18.5 years", standard deviation should be added.

"Conditional relative weight and conditional length/height data across infancy, childhood and adolescence were available for 957 participants..." Add percentage of the available data.

As mentioned in the methods, a flowchart might help understanding how from 1460 initial children (as for the method section), the authors only have measures for 957.

Table 1: add unit of measures for BMI and waist circumference

"...Mean outcome values of the main analysis samples were compared with mean outcome values of all participants...". The authors are comparing the outcomes variables of those included in the analysis with the outcome variables of the whole population (hence they are including the people in the analysis in both comparison groups). Suggest comparing the characteristics of those included in the sample with those that had to be excluded to look whether there are differences between the two groups.

Authors should be consistent in how ages are presented. Suggest choosing between the one between "4-11y" or "4 to 11 years" (similar for other ages) and be consistent through the paper.

Discussion:
"...showing that excessive weight gain from mid-childhood onwards, specifically after the second year of life ..." Wouldn't the second year of life refer to early-childhood in this study? Early childhood was previously defined 1-4 years.

"...Weight gain from birth to 4 years and lipid profile levels at 18 years was studied in the 1982 Cohort..." should read "were studied".

	“...with less consistent associations in girls compared to boys...” The authors should clarify this statement. “...the association with SBP became insignificant adjusted for current BMI...”. Current BMI is the one measured at 15 years? Authors should think on rephrasing this as “...after adjustment for BMI at 15 years”. course life should read life course. “For this reason, our findings support the initiative of improving nutrition during the "first 1000 days of life" (from conception up to age 2 years)...” As the authors pointed out, there is no anthropometry collected at two years of age. Hence, the authors cannot prove this conclusion based on what they have available. “...Given that cardiometabolic risk can track from adolescence to adulthood...” provide references for this statement.
--	--

REVIEWER	Fawaz Mzayek University of Memphis. USA
REVIEW RETURNED	22-Sep-2017

GENERAL COMMENTS	Growth across life course and cardiometabolic risk markers in 18 years old adolescents: the 1993 Pelotas Birth Cohort The manuscript examines the effect of childhood growth trajectories, across different age intervals, on hemodynamic and metabolic CV risk factors at 18 years of age. The study addresses an interesting topic from public health point of view because early-life interventions may reduce risk of chronic diseases according to the early origins of adult diseases hypothesis. At its current state the manuscript have several problems that need to be addressed. One important concern is the unclear relevance of the analysis of length/height for informing public health interventions—an objective that is stated in the introduction and the discussion—as height is not a modifiable variable. Other important points are included here, with additional feedback provided on the manuscript text. Methods: 1- The description of the parent study was provided as part of the current study, with mentioning of a large sample size and very high retention rates (page 6, lines 25-39). This is confusing and potentially misleading. The current study uses data of no more than 957 participants, yet this number is nowhere to be found in the method section. 2- The analysis did not adjust for important confounders, such as maternal smoking, breastfeeding, participant’s smoking. The authors failed to mention this limitation. Also, no description was provided on how important comorbidities, such as type 1 diabetes, were handled. 3- This reviewer is not expert in the statistical analysis that was used, but suspect that the conditional variables used in the regression models are correlated, because they are adjusted residuals.
---

	A better approach could be to perform hierarchical analysis, where the repeated growth variables are nested within the individual. This approach would still use all the information in the data, simplify the reporting and the interpretation of the findings (only one coefficient for the trajectory of growth change, instead of five), and, consequently, will not inflate the type-I error. 4- The authors are commended for providing a detailed comparison of the study subsample with the original population to determine whether there is evidence of selection bias. Discussion: 1- While detailed description of the results is the norm in the “Results” section, the discussion should summarize the findings to provide a meaningful message for the readers. From Tables 1 and 2, it can be seen that growth is associated with SBP, BMI and WC in both sexes, and with LDL-C in older boys and HDL-C (inversely) in older girls. The statement: “We observed that higher conditional relative weight at most age periods in childhood (1-4 y and 4-11 y) and adolescence (11-15 y and 15-18 y) was positively associated with most of the cardiometabolic markers (CRP, lipid profile, SBP, DBP, BMI and WC) at 18 years old adolescents of both sexes...” is vague and not very informative. As for the effect of height, no message can be gleaned, especially given the inconsistent results and the inflated type-I error. 2- Page 16, lines 27-30: this statement is not supported by the fact that only a small subgroup is used in this study 3- Page 16, lines 46-55: again, the retention rates and data completeness of the parent study are not relevant to this manuscript and stating them when discussing a limitation of the study can be misleading. 4- It is hard to evaluate the clinical significance of the reported associations since standardized beta does not reflect the original measurement’s unit. It would be helpful if the authors can provided some idea about the clinical effect of the change in weight (e.g., the effect of 1 unite change in weight on SBP). However, I recognize that this may be difficult to do due to the way the analysis was performed.
--	---

REVIEWER	Madhumita Sinha, MD National Institutes of Health, USA
REVIEW RETURNED	26-Sep-2017

GENERAL COMMENTS	This study evaluated the association between growth at different time periods from birth through childhood and adolescence and cardiometabolic risk markers at 18 years to identify periods of vulnerability by using data from the longitudinal Pelotas birth cohort. This is an important study. The biochemical outcome measures include an inflammatory marker (CRP), lipid profile, BMI and waist circumference. Comment: 1. Although the study emphasizes cardiometabolic risk, the outcomes measured focus primarily on cardiovascular risk markers that include CRP and lipid profile and blood pressure. It does not include any metabolic risk variables. Do the authors have data on any glycemic variables? Fasting plasma glucose, 2-hour plasma glucose or HbA1c?
---

	2. The authors have considered systolic and diastolic blood pressures separately, it would be interesting to see the trends for Mean Arterial Pressure (MAP) calculated easily from the systolic and diastolic BP, it is a useful measure since it reflects overall blood to the vital organs and is a good indicator of perfusion pressure. 3. From table 1 it seems with a mean BMI of 23 (M) and 23.58 (F) this is a normal weight youth population. Is it possible to classify weight categories among the different age groups (1-4, 4-11, 11-15, 15-18) using WHO weight for length/BMI based age-sex percentiles or z-scores? This may be important since the comparison is with sex-specific internal z-scores. How does this population compare to the rest of the Brazilian pediatric population? 4. It has also been mentioned that for the youth, weight was measured using a scale coupled to a BodPod equipment, do the authors then have a access to additional body composition measures? Fat mass? Fat free mass? 5. In absence of any metabolic risk measures the authors do mention in their conclusion "This reinforces efforts....after their first 2 years of life for cardiovascular prevention" (page 17 last para), I feel this study is primarily looking at periods of accelerated growth and its association with cardiovascular risk measures only.
--	--

VERSION 1 – AUTHOR RESPONSE

Reviewer: 1

Reviewer Name: Chiara Di Gravio

Institution and Country: MRC Lifecourse Epidemiology Unit, University of Southampton, United Kingdom

Please state any competing interests: None Declared

Please leave your comments for the authors below

Comment: The author looked at associations between growth trajectories from birth to adolescence and cardiometabolic risk markers at 18 years. Higher weight gain was positively associated with cardiometabolic markers, some associations between linear growth and cardiometabolic risk markers were also found. The statistical analysis is well-explained. Below are some comments/suggestions for the authors:

Author response: We thank the reviewer and appreciate the opportunity to respond to her comments, which we believe assisted us in making numerous improvements to our manuscript.

Abstract:

If space permit, include relevant estimates and 95% confidence interval in the results section and define TGL in the primary outcome section.

Author response: The maximum word count in the text does not permit the inclusion of statistics, however, all estimates and 95% confidence intervals appear in Tables 2 and 3. TGL has now been defined (triglycerides = TGL) in the outcome section.

Introduction:

“...However, there is controversy about which age intervals of accelerated growth (weight and height changes) lead to the development of chronic conditions...”

Which age intervals are considered in the literature and what controversy does the authors refer to? This statement needs more detail and clarification.

Author response: While some studies has offered evidence of rapid weight gain during the first year of life as a risk factor for cardiometabolic diseases, others have shown that the most harmful period for cardiometabolic risk was mid-childhood (6 to 8 years old). Therefore, a controversy remains regarding which age intervals of rapid weight gain should be paid special attention to prevent future diseases. We have now explained this further in the text and added three references [5, 7, 8]: Bird, 2005; Fisher, 2006; Eriksson, 2011.

Comment: “Dissimilar consequences of these measures have been found...”

Provide references supporting this statement.

Author response: Two references [12, 13] have now been added: Adair, 2013 and Victora, 2008.

Methods:

In how many hospitals did women delivered? Were all the hospitals similar in term of population they cater to?

Author response: The total number of hospitals in the city of Pelotas is five and they cater to the total population of Pelotas (99% of all city births occurs in this five hospitals). The five hospitals were included in the study. This is now explained in the Methods (first paragraph, Methods section).

Comment: It is unclear whether the 1,460 children mentioned are all of the children followed-up at 1 and 4 years, or if they are the random sample of 20% of the remaining population. Suggest having different Ns in the text, one indicating how many low birth weight children were included, the other referring to the random sample of 20% of the remaining population.

Author response: The target population for 1- and 4-years follow-ups was a subsample (N= 1,460 children) of the original cohort consisting of all children born low birth weight (n=510) plus a random sample of 20% of children who were not born low birth weight (n= 950).

A sentence to explain this has now been added to the Methods section (paragraph 2, Methods section).

Comment: Are the children randomly selected and included in the 1 year follow up, the same children studied at the 4 year follow up? Or was the sample randomly selected at both visit separately? This information becomes essential as the statistical methodology used (conditionals) is strongly affected by missing values.

Author response: Children selected at the 1 year follow-up were the same as those assessed at the 4 year follow-up. At the 4-years follow-up, 93.4% of the target population were identified (N=1273). We have now rephrased the second paragraph of the Methods section to make this clearer and a new reference has also been added [15] which provides a detailed explanation of every follow-up of the 1993 Pelotas Birth Cohort (Victora, 2006).

Comment: Suggest adding the number of available measure at each follow up (together with the response rate already in the text). A flowchart of the cohort might help the reader in understanding the sample size at each follow-up.

Author response: The samples size of every follow-up used in this study have been added along with the corresponding response rates. A diagram with the description of the cohort follow-ups has been added as Supplementary material (Supplementary file 1).

Assessment of outcomes:

“The tests were not taken in pregnant or suspected pregnant.” How many women?

Author response: The pregnant or suspected pregnant participants at the moment of the follow-up were 59 girls, which has been now added in the Methods section (paragraph 2, Assessment of outcomes subsection).

Statistical Analyses:

Why using geometric means for CRP and TGL?

Author response: Geometric means were used to describe CRP and TGL because the distribution of these variables is asymmetric. This information has now been provided when describing statistical analyses (paragraph 1, Statistical analyses subsection).

Comment: Why were the analysis stratified by sex?

Author response: We found evidence of interaction by sex in most of associations assessed, therefore, we showed analyses stratified by sex. We have now added a sentence in the Methods section that explains this (paragraph 1, Statistical analyses subsection).

Results:

“... a mean age of 18.5 years”, standard deviation should be added.

Author response: SD has now been added (paragraph1, Results section).

Comment: “Conditional relative weight and conditional length/height data across infancy, childhood and adolescence were available for 957 participants...” Add percentage of the available data.

Author response: The percentage of the main analysis sample size (N=957) in relation to the total number of cohort members followed-up at 18 years old (N=4106) has now been added in the text (paragraph 1, Results section).

Comment: As mentioned in the methods, a flowchart might help understanding how from 1460 initial children (as for the method section), the authors only have measures for 957.

Author response: We have conditional growth measures only for 957 cohort members because of losses to follow-up. The target sample size intended to be evaluated at 1- and 4-years follow-ups were 1460 participants . Of these, 1363 and 1273 accepted to participate, respectively. In addition, to create the conditional growth variables, we also used weight and height data from the perinatal, 11-, 15- and 18-years old follow-ups, in which the numbers of participants evaluated were 5249, 4452, 4349 and 4106, respectively. Thus, the total number of participants with complete weight and length/height data from all follow-ups to create the conditional growth measures was 957 individuals. This total was reduced to 917 individuals for outcomes that involved collection of blood samples and 946 individuals for blood pressure, BMI and WC, due to some refuses during physical evaluation. A flowchart has been added to better explained the sample sizes (Supplementary file 1).

Table 1: add unit of measures for BMI and waist circumference

Author response: The units of measures were added.

Comment: "...Mean outcome values of the main analysis samples were compared with mean outcome values of all participants...". The authors are comparing the outcomes variables of those included in the analysis with the outcome variables of the whole population (hence they are including the people in the analysis in both comparison groups). Suggest comparing the characteristics of those included in the sample with those that had to be excluded to look whether there are differences between the two groups.

Author response: Point taken. A new table (Supplementary file 3) has been added comparing the outcome between participants included in the analysis with those excluded. Only few divergences were observed with the previous table comparing those included in the analysis with the whole population. In the new analyses comparing included and excluded participants, we do not find statistical evidence of differences in mean values of C-reactive protein in boys. However, we found statistical evidence of differences in mean of triglycerides and waist circumference among boys.

Comment: Authors should be consistent in how ages are presented. Suggest choosing between the one between "4-11y" or "4 to 11 years" (similar for other ages) and be consistent through the paper.

Author response: Agreed. The manuscript was revised and age intervals have been rewritten in order to be consistent through the text as: x to x years.

Discussion:

"...showing that excessive weight gain from mid-childhood onwards, specifically after the second year of life ...". Wouldn't the second year of life refer to early-childhood in this study? Early childhood was previously defined 1-4 years.

Author response: Agreed. The sentence has been rephrased (paragraph 2, Discussion section).

Comment: "...Weight gain from birth to 4 years and lipid profile levels at 18 years was studied in the 1982 Cohort..." should read "were studied".

Author response: Agreed (paragraph 2, Discussion section).

Comment: "...with less consistent associations in girls compared to boys..." The authors should clarify this statement.

Author response: We have now deleted this statement as this made part of a previous version of this manuscript.

Comment: "...the association with SBP became insignificant adjusted for current BMI...". Current BMI is the one measured at 15 years? Authors should think on rephrasing this as "...after adjustment for BMI at 15 years".

Author response: Agreed. The sentence has been rephrased according reviewer's suggestion (paragraph 5, Discussion section).

Comment: course life should read life course.

Author response: Agreed (paragraph5, Discussion section).

Comment: "For this reason, our findings support the initiative of improving nutrition during the "first 1000 days of life" (from conception up to age 2 years)..." As the authors pointed out, there is no anthropometry collected at two years of age. Hence, the authors cannot prove this conclusion based on what they have available.

Author response: Agreed. We have now rephrased this sentence (paragraph 7, Discussion section).

Comment: "...Given that cardiometabolic risk can track from adolescence to adulthood..." provide references for this statement.

Author response: Two references [29, 30] have now been added to the statement (Camhi, 2010; Harding, 2016).

Reviewer: 2

Reviewer Name: Fawaz Mzayek

Institution and Country: University of Memphis. USA

Please state any competing interests: None declared

Please leave your comments for the authors below

Comment: Growth across life course and cardiometabolic risk markers in 18 years old adolescents: the 1993 Pelotas Birth Cohort

The manuscript examines the effect of childhood growth trajectories, across different age intervals, on hemodynamic and metabolic CV risk factors at 18 years of age. The study addresses an interesting topic from public health point of view because early-life interventions may reduce risk of chronic diseases according to the early origins of adult diseases hypothesis.

At its current state the manuscript have several problems that need to be addressed. One important concern is the unclear relevance of the analysis of length/height for informing public health interventions—an objective that is stated in the introduction and the discussion—as height is not a modifiable variable. Other important points are included here, with additional feedback provided on the manuscript text.

Author response: We thank the reviewer for highlighting the importance of our work.

Regarding the comment on relevance of length/height analysis, we would like to emphasize that one of the main message of our study is that provides evidence of dissimilar associations between weight gain and linear growth with cardiovascular risk factors. While relative weight gain was associated with most of cardiovascular outcomes at ages 1, 4, 11, 15; conditional length/height were not associated. However, faster linear growth at age 2 year and mid-childhood has been associated with a reduced risk of short adult stature and of not completing secondary school and increased likelihood of overweight and elevated blood pressure in other studies. Therefore, further studies to elucidate on the role of linear growth on cardiometabolic risk in order to inform adequate public health interventions are needed.

Author responses for the reviewer comments provided on the manuscript text:

- Comment 1: Not sure whether a limitation needs to be stated in the abstract: Agreed.
- Comment 2: Inaccurate. By definition, BMI is a measure of weight accounting for height: In the sentence "Studies on growth trajectories throughout life course and adult outcomes examine weight gain, without any distinction between weight gain relative to height and linear growth", we did not mention BMI, we are referring in this sentence to weight gain.
- Comment 3: Are these fasting levels?: We have now rephrased the sentence to clarify this. Please see answer to Comment 4.
- Comment 4: What is meant by "random blood samples": The sentence was rephrased to clarify what we meant with random blood samples, which refers to venous blood samples at any time of the day, independently of fasting status.
- Comment 5: This is not a standard way to measure WC: We acknowledge that exist various standardized procedures to measure waist circumference. In the Pelotas Birth Cohorts, we maintained the same waist circumference measure over the years. We added the reference to the measurement (Ref. 17: Lohman, 1988).
- Comment 6: Interval (?): Line 29, page 7, we did mean "internal", as the transformation of weight and height variables into z-scores was based on the own cohort members measurements. To make it clearer, we have now rephrased this as "sex-specific internally derived z-scores".
- Comment 7: Since these numbers where not used in sample description (i.e., Table 1), they should be deleted because they may confuse the reader: Agreed.

Methods:

1- The description of the parent study was provided as part of the current study, with mentioning of a large sample size and very high retention rates (page 6, lines 25-39). This is confusing and potentially misleading. The current study uses data of no more than 957 participants, yet this number is nowhere to be found in the method section.

Author response: 957 is the number of cohort members with complete anthropometric information for conditional growth. The original Pelotas Birth Cohort comprised about 5000 newborns. However, subsamples of participants were evaluated at 1 and 4 years old of age. N= 1460 was the sample size intended to be evaluated at these follow-ups. Of these, 1363 and 1273 accepted to participate in this study, respectively. We also used weight and height data from the perinatal, 11- 15- and 18-years old follow-ups, in which the numbers of participants evaluated were 5249, 4452, 4349 and 4106, respectively. The total number of participants with weight and length/height data necessary to create the conditional growth measures was 957. The final analysis samples were 917 for outcomes with blood exams and 946 for blood pressure, BMI and WC, due to some losses during physical evaluation.

As a suggestion of Reviewer #1, a flowchart explaining all the phases of the 1993 Pelotas Birth Cohort has been added (Supplementary file 1).

2- The analysis did not adjust for important confounders, such as maternal smoking, breastfeeding, participant's smoking. The authors failed to mention this limitation. Also, no description was provided on how important comorbidities, such as type 1 diabetes, were handled.

Author response: Point taken. We have now repeated our analyses including breastfeeding (total duration of breastfeeding in months) and maternal smoking during pregnancy. This information was added to the Statistical analyses subsection. Tables 2 and 3 were revised accordingly the inclusion of the new confounders. The new adjustment did not change the results.

We have not included participant's smoking in our analyses as this would be a mediator in the association between conditional growth and cardiovascular risk factors.

The aim of the study was to assess the association between growth patterns from infancy to childhood and cardiovascular risk markers in adolescence, thus, only adjustment for potential confounders (variables that were at the highest level in the hierarchical model) was taken into account as suggested for several authors (Victora et al, 1997).

Victora CG, Huttly SR, Fuchs SC, Olinto MT. The role of conceptual frameworks in epidemiological analysis: a hierarchical approach. *International journal of epidemiology*. 1997 Feb 1;26(1):224-7.

3- This reviewer is not expert in the statistical analysis that was used, but suspect that the conditional variables used in the regression models are correlated, because they are adjusted residuals. A better approach could be to perform hierarchical analysis, where the repeated growth variables are nested within the individual. This approach would still use all the information in the data, simplify the reporting and the interpretation of the findings (only one coefficient for the trajectory of growth change, instead of five), and, consequently, will not inflate the type-I error.

Author response: The statistical method used to create the conditional variables is appropriate to overcome the problem of correlated measures over the time. The standardized residuals are independent. We were interested in assessing the associations between every age interval with the cardiovascular outcomes at age 18 years as the aim of our study was to examine the potential differences between each age interval. This approach has been used elsewhere (Adair, 2013; De França, 2016; Horta, 2017; de Mola, 2017; da Silva, 2015)

Adair LS, Fall CH, Osmond C, Stein AD, Martorell R, Ramirez-Zea M, et al. Associations of linear growth and relative weight gain during early life with adult health and human capital in countries of low and middle income: findings from five birth cohort studies. *Lancet*. 2013;382(9891):525-34. Epub 2013/04/02

De França GA, Rolfe EDL, Horta B, Gigante D, Yudkin J, Ong K, et al. Associations of birth weight, linear growth and relative weight gain throughout life with abdominal fat depots in adulthood: the 1982 Pelotas (Brazil) birth cohort study. *International journal of obesity (2005)*. 2016;40(1):14.

Horta BL, Victora CG, de Mola CL, Quevedo L, Pinheiro RT, Gigante DP, et al. Associations of linear growth and relative weight gain in early life with human capital at 30 years of age. *The Journal of pediatrics*. 2017;182:85-91. e3.

de Mola CL, de Avila Quevedo L, Pinheiro RT, Gonçalves H, Gigante DP, dos Santos Motta JV, et al. The Effect of Fetal and Childhood Growth over Depression in Early Adulthood in a Southern Brazilian Birth Cohort. *Plos One*. 2015;10(10):e0140621.

da Silva Linhares R, Gigante DP, de Barros FCLF, Horta BL. Carotid intima-media thickness at age 30, birth weight, accelerated growth during infancy and breastfeeding: a birth cohort study in Southern Brazil. *Plos One*. 2015;10(1):e0115166.

4- The authors are commended for providing a detailed comparison of the study subsample with the original population to determine whether there is evidence of selection bias.

Author response: We thank the reviewer for this comment.

Discussion:

1- While detailed description of the results is the norm in the "Results" section, the discussion should summarize the findings to provide a meaningful message for the readers. From Tables 1 and 2, it can be seen that growth is associated with SBP, BMI and WC in both sexes, and with LDL-C in older boys and HDL-C (inversely) in older girls. The statement: "We observed that higher conditional relative weight at most age periods in childhood (1-4 y and 4-11 y) and adolescence (11-15 y and 15-18 y) was positively associated with most of the cardiometabolic markers (CRP, lipid profile, SBP, DBP, BMI and WC) at 18 years old adolescents of both sexes..." is vague and not very informative. As for the effect of height, no message can be gleaned, especially given the inconsistent results and the inflated type-I error.

Author response: We have now rephrased this statement to be more precisely (paragraph 1, Discussion section).

2- Page 16, lines 27-30: this statement is not supported by the fact that only a small subgroup is used in this study

Author response: Agreed. The statement has been rephrased (paragraph 6, Discussion section).

3- Page 16, lines 46-55: again, the retention rates and data completeness of the parent study are not relevant to this manuscript and stating them when discussing a limitation of the study can be misleading.

Author response: Agreed. The paragraph of limitations has been rephrased (paragraph 7, Discussion section).

4- It is hard to evaluate the clinical significance of the reported associations since standardized beta does not reflect the original measurement's unit. It would be helpful if the authors can provided some idea about the clinical effect of the change in weight (e.g., the effect of 1 unite change in weight on SBP). However, I recognize that this may be difficult to do due to the way the analysis was performed.

Author response: Results (regression coefficients) are presented in original units (i.e., mg/dl or cm) in Supplementary file 5.

Reviewer: 3

Reviewer Name: Madhumita Sinha, MD

Institution and Country: National Institutes of Health, USA

Please state any competing interests: None

Please leave your comments for the authors below

Comment: This study evaluated the association between growth at different time periods from birth through childhood and adolescence and cardiometabolic risk markers at 18 years to identify periods of vulnerability by using data from the longitudinal Pelotas birth cohort. This is an important study.

The biochemical outcome measures include an inflammatory marker (CRP), lipid profile, BMI and waist circumference.

Response: We thank the reviewer for highlighting the importance of our work.

Comment:

1. Although the study emphasizes cardiometabolic risk, the outcomes measured focus primarily on cardiovascular risk markers that include CRP and lipid profile and blood pressure. It does not include any metabolic risk variables. Do the authors have data on any glycemic variables? Fasting plasma glucose, 2-hour plasma glucose or HbA1c?

Author response: Agreed, We have now changed the term "cardiometabolic" was changed for "cardiovascular". Glycemic variables will make part of another manuscript that is underway.

2. The authors have considered systolic and diastolic blood pressures separately, it would be interesting to see the trends for Mean Arterial Pressure (MAP) calculated easily from the systolic and diastolic BP, it is a useful measure since it reflects overall blood to the vital organs and is a good indicator of perfusion pressure.

Author response: Agreed. We calculated MAP according to the formula = $[(2 \times \text{diastolic BP}) + \text{systolic BP}] / 3$. Results of the association between growth and MAP are shown in Supplementary file 4. In general, these results are consistent with those shown for systolic and diastolic BP in both sexes.

3. From table 1 it seems with a mean BMI of 23 (M) and 23.58 (F) this is a normal weight youth population. Is it possible to classify weight categories among the different age groups (1-4, 4-11, 11-15, 15-18) using WHO weight for length/BMI based age-sex percentiles or z-scores? This may be important since the comparison is with sex-specific internal z-scores. How does this population compare to the rest of the Brazilian pediatric population?

Author response: Conditional growth (conditional relative weight and conditional height) was performed using the WHO based age-sex z-scores, thus, we did not observe differences with conditional growth based on the internally derived z-scores.

4. It has also been mentioned that for the youth, weight was measured using a scale coupled to a BodPod equipment, do the authors then have a access to additional body composition measures? Fat mass? Fat free mass?

Author response: Yes, the 1993 Pelotas Birth Cohort, and the other cohorts in Pelotas (1982 and 2004 Pelotas Birth Cohorts) have information on body composition measured by BodPod. Other colleagues are leading the work on the associations between conditional growth and body composition, which is underway.

5. In absence of any metabolic risk measures the authors do mention in their conclusion "This reinforces efforts...after their first 2 years of life for cardiovascular prevention" (page 17 last para), I feel this study is primarily looking at periods of accelerated growth and its association with cardiovascular risk measures only.

Author response: Agreed. The term "cardiometabolic" was changed for "cardiovascular".

VERSION 2 – REVIEW

REVIEWER	Chiara Di Gravio MRC Lifecourse Epidemiology Unit University of Southampton United Kingdom
REVIEW RETURNED	04-Nov-2017

GENERAL COMMENTS	The authors have addressed all my previous comments. I have no further comments to make
---

REVIEWER	Madhumita Sinha National Institutes of Health
REVIEW RETURNED	06-Nov-2017

GENERAL COMMENTS	Satisfied with revisions
--------------------------